# Effect of Phosphorus Supply Levels on Nodule Nitrogen Fixation and Nitrogen Accumulation in Soybean (*Glycine max* L.)

**Hongyu Li** [1], **Lihong Wang** [2], **Zuowei Zhang** [2,*], **Aizheng Yang** [2]  **and Deping Liu** [2]

1   College of Agriculture, Northeast Agricultural University, Harbin 150030, China
2   School of Water Conservancy and Civil Engineering, Northeast Agricultural University, Harbin 150030, China
*   Correspondence: zzw@neau.edu.cn

**Abstract:** The specific mechanism by which phosphorus affects nodule nitrogen fixation and nitrogen absorption in soybeans remains inconclusive. To further quantitatively analyze the effect of phosphorus on nodule nitrogen fixation and nitrogen accumulation in soybeans, this experiment was carried out under sand culture conditions. The experiment consisted of six phosphorus supply levels (1 mg/L, 11 mg/L, 21 mg/L, 31 mg/L, 41 mg/L, 51 mg/L). The acetylene reduction method and $^{15}N$ tracer method (50 mg/L $(NH_4)_2SO_4$) were used to determine and analyze the nodule growth status, nodule nitrogenase activity, nitrogen content, and nodule nitrogen fixation rate at initial flowering (R1 stage), initial pod (R3 stage), seed filling (R5 stage) and maturity stages (R8 stage). The results are described as follows: 1. The nitrogen fixation of soybean nodules at different growth stages has different requirements for phosphorus supply levels. The initial flowering stage and seed-filling stage were 31 mg/L–41 mg/L, and the initial pod stage was 51 mg/L. 2. The nitrogen source in different parts of soybean showed different trends with different growth periods and phosphorus supply concentrations. Among them, from the initial flowering stage to the seed filling stage, the main body of the nitrogen supply of soybean shoots in the low phosphorus treatment (1 mg/L–31 mg/L) gradually changed from fertilizer nitrogen to nodule nitrogen fixation, while the main body of the nitrogen supply of soybean shoots in the high phosphorus treatment (41 mg/L–51 mg/L) always showed nodule nitrogen fixation and was transformed into fertilizer nitrogen at the mature stage. The main nitrogen supply to the roots of soybean at different levels of phosphorus supply from the initial flowering to the initial pods and maturity stage was fertilizer nitrogen, and the main nitrogen supply at the seed filling stage was nodule nitrogen fixation. The nitrogen supply to the main body of soybean nodules was constantly nodule nitrogen fixation. 3. Different phosphorus supply levels significantly affected the nitrogen fixation of soybean nodules ($R^2 \geq 0.803$), and both the acetylene reduction method and the $^{15}N$ tracer method could be used to determine the nitrogen fixation capacity of soybean nodules. This study indicated the optimal phosphorus supply level of nodules in different growth stages of soybean and clarified the main body of phosphorus supply in different parts of soybean at different growth stages, which pointed out the direction for further improving the utilization efficiency of soybean nitrogen and phosphorus fertilizer.

**Keywords:** phosphorus supply levels; nodulation; nitrogen accumulation; nitrogen fixation

## 1. Introduction

Phosphorus is one of the three basic elements required for crop growth, second only to nitrogen, which has a significant effect on the growth and nodulation process of leguminous crops [1–3]. According to statistics, 40% of the world's arable land grows crops whose yield is limited by phosphorus [4]. Studies have shown that the reasonable application of phosphorus fertilizer can regulate the growth and development of soybean nodules, enhance the nitrogen fixation capacity of nodules, and promote plant growth, thereby achieving the effect of phosphorus promoting nitrogen and increasing yield [5–11]. At present, the effects of phosphorus on nitrogen fixation in nodules mainly focus on the effect

of different phosphorus levels on nodule growth and metabolism [12–14]. Some studies have suggested that low phosphorus concentration reduces the nitrogenase activity in nodules of legume crops due to the reduction of ATP energy in nodules [15], the decrease in leghemoglobin content [16], decreased Fe content [5], and excessive secretion of organic acids [17]. Other studies have found that leguminous crops adapt to low phosphorus concentrations by increasing phytase and phosphatase activity in nodules [18,19]. Some studies have also suggested that low phosphorus concentration inhibits the nitrogen fixation of leguminous nodules and the nitrogen absorption of plants [20,21]. Additional analysis of the physiological effects of phosphorus levels on nitrogen fixation and metabolism in nodules showed that the effect of phosphorus on nitrogen fixation in nodules was achieved by regulating nitrogen metabolism [22]. Characterized by low phosphorus conditions, the proportion of nodule nitrogen accumulated by legume crops decreased significantly, while the nitrogen content in plants and leaves increased, indicating that the assimilated nitrogen had exceeded the demand of plant growth and development for nitrogen, and the accumulation of asparagine in roots and nodules increased, further indicating that the effect of phosphorus on nodule nitrogen fixation was achieved by regulating nitrogen metabolism [12–14]. The above studies have shown that phosphorus levels have a significant effect on nitrogen fixation and nitrogen uptake by nodules of leguminous crops, but they have not been able to quantitatively characterize the main nitrogen supply of leguminous crops in each growth period under different phosphorus supply levels, nor have they affected different growth stages according to the growth process. An in-depth analysis of the effect of nitrogen fixation by soybean nodules under the phosphorus supply level is not conducive to improving the utilization efficiency of nitrogen and phosphorus fertilizers in soybean. Therefore, this study intended to quantitatively characterize the nodule growth status, nodule nitrogenase activity, nodule nitrogen fixation rate, and nitrogen accumulation of soybean plants with different phosphorus supply levels under sand culture conditions by the $^{15}$N tracer method and provide a theoretical basis for analyzing the physiological mechanism of phosphorus application affecting soybean nodule nitrogen fixation and nitrogen absorption.

## 2. Materials and Methods

### 2.1. Test Site Overview

The study was conducted at the Experimental Base of Northeast Agricultural University in 2019, which is located in the Xiangfang District, Harbin City, Heilongjiang Province, China, with geographical coordinates of 126.43°43′ E, 45°44′ N. The annual precipitation is 500–550 mm, and the accumulated temperature is $\geq$10 °C and 2700 °C.

### 2.2. Test Materials

The soybean variety tested was Kenfeng 16 (*Glycine max* L.) (Heilongjiang Academy of Land Agricultural Reclamation Science, Heilongjiang, China). Labeled ammonium sulfate $^{15}$N (3.36% abundance) was purchased from Shanghai Chemical Industry Research Institute, and the tested phosphorus source was $KH_2PO_4$.

### 2.3. Experimental Design

Six phosphorus levels were set up in the experiment: 1 mg/L, 11 mg/L, 21 mg/L, 31 mg/L, 41 mg/L, and 51 mg/L, expressed as P1, P11, P21, P31, P41, and P51, respectively. In each treatment, $(^{15}NH_4)_2SO_4$ with a $^{15}$N abundance of 3.36% was used as a nitrogen source to analyze the nitrogen fixation ability of nodules. Phosphorus levels and the composition of the nutrient solution refer to Yao et al. [23] and Li et al. [24] using the nutrient solution formula, with a slight improvement. The specific components are shown in Tables S1 and S2.



### 2.4. Test Treatment

Potted cultivation with river sand was used in this study. Each plastic pot was 0.30 m in diameter and 0.28 m in height. Two drainage holes 1 cm in diameter were drilled at the bottom of the pot, with one on each side of the partition plate. Each pot was filled with 20 kg of washed sand. Before use, the sand was thoroughly washed with tap water and then rinsed twice with distilled water. The soybean seeds were sown in fine sand at a depth of 2 cm and cultured in an incubator at 30 °C for approximately 3 days. When the distance between the growing point of the cotyledon and the tip of the root reached approximately 7–10 cm, the root system of soybean seedlings was exposed by rinsing with distilled water; then, uniform seedlings were selected for transplanting, and two seedlings were kept per pot.

The inoculation of rhizobia was carried out when the opposite true leaves of soybean were fully expanded. The soybean nodules stored in the field in the previous year were ground into powder and then added to the nutrient solution, and each liter of nutrient solution contained about 5 g nodules for continuous inoculation for 5 days. According to the characteristics of soybean fertilizer demand, the nutrient supply was divided into three periods. The first period was from transplanting the seedlings before the VC stage (the unfolded cotyledon stage), and they were drenched with distilled water once a day, 500 mL per pot. The second period was the unfolded cotyledon to fully expanded stage to the $V_4$ stage (fourth trifoliate leaf stage), and the basic nutrient solution of 50 mg/L $(NH_4)_2SO_4$ and 31 mg/L $KH_2PO_4$ was poured once a day, 500 mL each time. The third period was from V4 to harvest, and $^{15}N$ labeling and different phosphorus supply levels were carried out. This period is divided into three stages, of which the V4 stage to the R1 stage (initial flowering stage) is the first stage, and the nutrient solution is poured once a day, with 500 mL each time. The second stage is from the R1 stage to the R8 stage (maturity stage), and the same nutrition is used as the first stage, but the nutrient solution is poured twice a day, 500 mL each time, that is, 1000 mL of nutrients per pot every day. From the R8 stage to harvest is the third stage, considering the decline in nutrients required by plants, the supply of nutrient solution was reduced to 500 mL once a day (Table S3).

### 2.5. Sampling and Measurement

Samples were taken 4 times at initial flowering (R1 stage, 42 days), initial pod (R3 stage, 15 days), seed filling (R5 stage, 16 days), and maturity stages (R8 stage, 38 days)., and each treatment was repeated 4 times. The shoots were cut along the boundary between the ground and the ground at 8:00–10:00 AM on a sunny day, and the underground part was washed with distilled water to remove the sand and blotted dry with filter paper, and then the nodule nitrogenase activity was measured [25]. After the measurement, the nodules were removed and recorded. The number of nodules and the dry weight of nodules and roots were determined after drying at 60 °C. The shoots were fixed at 105 °C for 1 h, dried at 65 °C to constant weight, and weighed. After crushing, the $^{15}N$ abundance and plant nitrogen content of each part were determined. When soybean leaves turned yellow, to reduce the error of dry matter quality caused by falling leaves, the fallen leaves were picked up twice a day, morning and evening, and collected in mesh bags for the calculation of soybean dry matter accumulation in the R8 period.

The nitrogenase activity of nodules was determined by Acetylene reduction activity according to the method of Gremaud and Harper (1986).

Plant nitrogen content was determined by the B324 automatic Kjeldahl nitrogen analyzer produced by Shanghai Shengsheng Automatic Analytical Instrument Co., LTD., China.

$^{15}N$ abundance was determined by a mass spectrometer using a double-channel (DI) measurement (Thermo-Fisher Delta V Advantage IRMS, Waltham, MA, USA).

### 2.6. Data Calculations

The sand culture was used in the experiment and there was no soil factor interference; therefore, the nitrogen source of the plant was derived from the application of [15]N-labeled fertilizer nitrogen and atmospheric nitrogen fixed by nodules.

The nitrogenase activity per unit nodule weight was calculated as follows:

$$\text{SNA} = \frac{E_R}{D_N - T_R} \tag{1}$$

where SNA is specific nitrogenase activity (unit is $C_2H_4$ µmol $g^{-1}$ Nodule Dry Mass $h^{-1)}$), $E_R$ is the amount of reduced ethylene (unit is µmol), $D_N$ is the dry weight of the nodule (unit is g), and $T_R$ is the reaction time (unit is h), the same as below.

The nitrogenase activity per plant was calculated as follows:

$$\text{ARA} = \frac{E_R}{T_R} \tag{2}$$

where ARA is acetylene reduction activity (unit is $C_2H_4$ µmol $h^{-1}$ $\text{Plant}^{-1)}$).

The percentage of nitrogen fixation originating from nodules in the plant was calculated as follows:

$$\text{RNNF} = \frac{f_{\text{fertilizer}} - f_{\text{treatment}}}{f_{\text{ertilizer}} - f_0} \tag{3}$$

The formula source: $\text{RNNF} \times f_0 + (1\text{-RNNF}) f_{\text{fertilizer}} = f_{\text{treatment}}$, where RNNF is nodule nitrogen fixation rate (unit is %), 1-RNNF is fertilizer utilization rate (unit is %), $f_0$ is background natural abundance (unit is %), and $f_{\text{treatment}}$ is treatment [15]N abundance (unit is %).

### 2.7. Statistical Analyses

All statistical analyses were performed using SPSS 21.0 (SPSS Inc., Chicago, IL, USA). All the data were subjected to a normality test prior to a one-way analysis of variance (ANOVA), and Duncan's multiple range test was used at a significance level of $p < 0.05$.

## 3. Results

### 3.1. Effect of Different Phosphorus Supply Levels on Nodulation and Nitrogen Fixation

Table 1 shows the number and dry weight of soybean nodules under different phosphorus supply levels from the initial flowering stage (R1) to the seed-filling stage (R5). From the perspective of the growth process, the dry weight and number of soybean nodules in each treatment showed different trends. When the phosphorus supply level increased from 1 mg/L to 31 mg/L, the number of soybean nodules showed a trend of increasing first and then decreasing with the progress of the growth process, and the peak value appeared in the initial pod stage (R3). When the phosphorus supply level was further increased to 41 mg/L and above, it showed a linear growth trend, and the peak appeared in the seed-filling stage (R5). The dry weight of nodules showed a gradually increasing trend with the advancement of the reproductive process. By comparing the number of nodules and the dry weight of nodules, it was found that the number of nodules decreased under the phosphorus supply level of 1 mg/L–31 mg/L from the initial pod stage to the seed filling stage, but the dry weight of nodules increased. From the perspective of phosphorus supply level, the number of nodules and dry weight of soybean nodules all increased with increasing phosphorus supply level. When the phosphorus supply level increased to 41 mg/L, the number of nodules and dry weight of soybean did not increase significantly when the phosphorus supply level was increased again.

**Table 1.** Effects of different phosphorus supply levels on soybean nodule number and nodule dry weight.

| | Treatments | R1 | R3 | R5 |
|---|---|---|---|---|
| Nodule Number (Per Plant) | P1 | 66 ± 8.25 c | 70 ± 10.53 c | 48 ± 2.65 c |
| | P11 | 67 ± 7.17 c | 79 ± 9.02 bc | 57 ± 0.88 c |
| | P21 | 70 ± 0.88 bc | 92 ± 13.97 bc | 66 ± 10.09 c |
| | P31 | 85 ± 4.63 b | 124 ± 13.92 bc | 120 ± 7.22 b |
| | P41 | 120 ± 2.89 a | 142 ± 7.22 ab | 191 ± 5.49 a |
| | P51 | 124 ± 2.03 a | 197 ± 17.59 a | 207 ± 10.39 a |
| Nodule Weight (g/Plant) | P1 | 0.10 ± 0.01 e | 0.12 ± 0.07 d | 0.13 ± 0.01 c |
| | P11 | 0.10 ± 0.01 e | 0.18 ± 0.01 d | 0.26 ± 0.06 c |
| | P21 | 0.15 ± 0.01 d | 0.52 ± 0.01 c | 0.72 ± 0.02 b |
| | P31 | 0.25 ± 0.01 c | 0.69 ± 0.15 bc | 1.02 ± 0.19 b |
| | P41 | 0.26 ± 0.01 b | 0.77 ± 0.05 ab | 1.41 ± 0.16 a |
| | P51 | 0.28 ± 0.01 a | 0.88 ± 0.07 a | 1.42 ± 0.06 a |

The data are represented as the mean values ± standard error and independent measurements with three replicates. Different lowercase letters indicate the differences between treatments at a significance level of 5%.

Table 2 shows soybean SNA and ARA under different phosphorus levels from the initial flowering stage (R1) to the seed-filling stage (R5). From the perspective of the growth process, SNA and ARA showed different trends. SNA showed a linear downward trend with the progress of the growth process at the phosphorus supply level of 1 mg/L-41 mg/L, while at the phosphorus supply level of 51 mg/L, SNA showed a trend of first increasing and then decreasing, and the maximum value appeared in the initial pod stage (R3). ARA showed a linear downward trend with the growth process under the phosphorus supply level of 1 mg/L, first increasing and then decreasing under the phosphorus supply levels of 11 mg/L and 51 mg/L, and the peak appeared at the initial pod stage (R3), while the phosphorus supply levels from 21 mg/L to 41 mg/L showed a linear upward trend. From the perspective of phosphorus supply level, SNA and ARA showed similar trends in the R1 and R5 stages; both showed a unimodal curve with the increase in phosphorus supply level, and the maximum appeared at the phosphorus supply level of 31 mg/L–41 mg/L. SNA and ARA in the R3 stage increased with the increase in phosphorus supply level, and the maximum values appeared at the 51 mg/L phosphorus supply level.

**Table 2.** Effects of different phosphorus supply levels on soybean SNA and ARA.

| | Treatments | R1 | R3 | R5 |
|---|---|---|---|---|
| SNA($C_2H_4$ µmol g$^{-1}$ Nodule Dry Mass h$^{-1}$) | P1 | 27.62 ± 1.81 d | 13.96 ± 0.82 e | 7.83 ± 0.23 d |
| | P11 | 45.63 ± 6.85 c | 31.72 ± 1.04 d | 22.25 ± 2.17 c |
| | P21 | 78.82 ± 3.97 ab | 33.74 ± 0.10 d | 33.08 ± 0.22 ab |
| | P31 | 92.04 ± 8.50 a | 42.01 ± 0.14 c | 34.97 ± 0.05 a |
| | P41 | 73.28 ± 3.43 b | 51.27 ± 6.39 b | 32.79 ± 0.40 ab |
| | P51 | 71.12 ± 1.51 b | 76.36 ± 0.62 a | 31.06 ± 1.80 b |
| ARA($C_2H_4$ umol h$^{-1}$ Plant$^{-1}$) | P1 | 3.00 ± 0.29 d | 1.35 ± 0.12 d | 0.86 ± 0.02 d |
| | P11 | 3.70 ± 0.50 d | 5.67 ± 0.39 d | 5.44 ± 0.55 d |
| | P21 | 11.96 ± 0.74 c | 17.60 ± 0.16 cd | 23.92 ± 0.77 c |
| | P31 | 22.63 ± 2.49 a | 29.13 ± 2.07 bc | 35.82 ± 0.47 b |
| | P41 | 18.79 ± 0.09 b | 39.80 ± 7.47 b | 46.07 ± 4.92 a |
| | P51 | 19.82 ± 0.43 ab | 66.89 ± 13.61 a | 44.30 ± 4.52 ab |

Different lowercase letters indicate the differences between treatments at a significance level of 5%.

### 3.2. Effect of Different Phosphorus Supply Levels on Soybean Nitrogen Absorption and Distribution

3.2.1. The Effect of Different Phosphorus Supply Levels on the Abundance of $^{15}$N in Soybean Plants

Table 3 shows that the abundance of $^{15}$N in soybean plants showed different changes with the progress of the growth process. The abundance of $^{15}$N in shoots and roots showed a decreasing trend from the R1-R5 stage and an increasing trend from the R5–R8 stage. Since $^{15}$N is mainly derived from the labeled fertilizer nitrogen, it means that from the initial flowering stage to the seed-filling stage, as the nodules grew and provided nitrogen,

the fertilizer nitrogen absorbed by the shoots and roots gradually became unable to meet their own needs, and an increasing amount of nitrogen was fixed through the nodules. To obtain nitrogen, as the nodules continue to decline from the seed-filling stage to the mature stage, the nitrogen absorbed by the shoots and roots comes from fertilizer nitrogen, which in turn causes the $^{15}$N abundance to rise in the R8 stage, completing a round of major nitrogen supply. It can also be seen from the data of different treatments in each growth period in the table that the abundance of $^{15}$N in each part of soybean increased first and then decreased with the increase in phosphorus supply level, and the phosphorus supply level peaked at 11 mg/L and then increased. The abundance of $^{15}$N at the phosphorus supply level was significantly reduced.

**Table 3.** $^{15}$N abundance (%) in plant organs of soybean under different phosphorus supply levels.

| Stages | Treatments | Shoot | Root | Nodule |
|---|---|---|---|---|
| R1 | P1 | 2.27 ± 0.01 b | 2.41 ± 0.01 b | 1.12 ± 0.03 a |
| | P11 | 2.33 ± 0.01 a | 2.54 ± 0.01 a | 1.15 ± 0.04 a |
| | P21 | 2.07 ± 0.01 c | 2.31 ± 0.01 c | 0.96 ± 0.03 b |
| | P31 | 1.97 ± 0.01 d | 2.28 ± 0.01 d | 0.94 ± 0.09 b |
| | P41 | 1.55 ± 0.01 f | 1.96 ± 0.01 f | 0.78 ± 0.02 c |
| | P51 | 1.65 ± 0.01 e | 1.97 ± 0.01 e | 0.87 ± 0.03 bc |
| R3 | P1 | 1.89 ± 0.01 c | 2.07 ± 0.01 c | 0.87 ± 0.01 abc |
| | P11 | 1.99 ± 0.01 a | 2.17 ± 0.01 a | 0.92 ± 0.01 a |
| | P21 | 1.91 ± 0.01 b | 2.14 ± 0.01 b | 0.90 ± 0.01 ab |
| | P31 | 1.74 ± 0.01 d | 2.04 ± 0.01 d | 0.81 ± 0.03 bc |
| | P41 | 1.74 ± 0.01 d | 2.04 ± 0.01 d | 0.83 ± 0.04 abc |
| | P51 | 1.50 ± 0.01 e | 1.86 ± 0.01 e | 0.77 ± 0.06 c |
| R5 | P1 | 1.38 ± 0.01 c | 1.39 ± 0.01 b | 0.67 ± 0.02 ab |
| | P11 | 1.46 ± 0.01 a | 1.45 ± 0.02 a | 0.69 ± 0.01 ab |
| | P21 | 1.41 ± 0.01 b | 1.33 ± 0.01 c | 0.68 ± 0.01 a |
| | P31 | 1.25 ± 0.01 d | 1.26 ± 0.01 d | 0.62 ± 0.01 b |
| | P41 | 1.19 ± 0.01 e | 1.18 ± 0.01 e | 0.65 ± 0.05 ab |
| | P51 | 1.18 ± 0.01 e | 1.15 ± 0.01 e | 0.66 ± 0.03 ab |
| R8 | P1 | 2.41 ± 0.04 a | 2.42 ± 0.02 a | - |
| | P11 | 2.53 ± 0.04 a | 2.49 ± 0.04 a | - |
| | P21 | 2.22 ± 0.05 b | 2.13 ± 0.01 b | - |
| | P31 | 1.95 ± 0.10 c | 1.94 ± 0.09 b | - |
| | P41 | 2.15 ± 0.05 b | 2.01 ± 0.17 b | - |
| | P51 | 2.11 ± 0.01 bc | 1.96 ± 0.05 b | - |

Different lowercase letters indicate the differences between treatments at a significance level of 5%.

### 3.2.2. Proportion of Nitrogen Sources in Various Organs of Soybean under Different Phosphorus Supply Levels

Table 4 shows the proportion of nitrogen sources in soybean organs under different phosphorus supply levels. It can be seen from the table that the sources of nitrogen in different parts of soybean showed different changes with different growth stages. From the R1 stage to the R5 stage, the proportion of fertilizer nitrogen absorbed by each part of the soybean showed a downward trend and increased from the R5 stage to the R8 stage (except nodules). The proportion of nitrogen fixation from nodules and the proportion of nitrogen absorbed from fertilizers in each part showed an opposite trend with the growth process, and the nitrogen source of each part of the soybean showed different changes with changes in the level of phosphorus supply. In addition, from Table 4, it can be seen that the nitrogen sources of different parts of soybean show different changing rules with the phosphorus supply level. Among them, in the R1 stage, the shoots showed that when the phosphorus supply level increased from 1 mg/L to 31 mg/L, the nitrogen source from fertilizer nitrogen accounted for 53.58–65.52% (>50%), and from a nodule, nitrogen fixation accounted for 34.48–46.42% (<50%). When the phosphorus supply level was 41 mg/L to 51 mg/L, the nitrogen source from fertilizer nitrogen accounted for 39.57–42.67% (<50%), and from nodule nitrogen fixation accounted for 57.33–60.43% (>50%). When the phosphorus supply level increased from 1 mg/L to 21 mg/L in the R3 stage, the nitrogen in

the shoots that came from fertilizer nitrogen accounted for 50.08–54.21%, and the nitrogen fixation from nodules accounted for 45.79–49.20%; however when the phosphorus supply level was 31 mg/L–51 mg/L, nitrogen from fertilizers accounted for 37.91–45.85%, and nitrogen from nodules accounted for 54.15–62.09%When the phosphorus supply level increased from 1 mg/L to 51 mg/L in the R5 stage, the shoot nitrogen from fertilizer nitrogen accounted for 27.18–36.50%, and the nitrogen fixation from nodules accounted for 63.50–72.81%. By comparing the changes in nitrogen sources in each treatment in the R1–R5 stages, it can be seen that the main nitrogen supply under low phosphorus treatment will gradually change from fertilizer nitrogen to nitrogen fixation by the nodules as the growth stage progresses. At the R8 stage, the nitrogen in the shoots was 52.68–68.22% from fertilizers and 27.65–47.32% from nodules, The nitrogen source of roots in each growth stage was as follows: when the phosphorus supply level increased from 1 mg/L to 51 mg/L at the R1 stage, the nitrogen source of fertilizer nitrogen accounted for 53.01–72.47%, and the nitrogen fixation from nodules accounted for 27.53–46.99%. During the R3 period, nitrogen from fertilizers accounted for 49.99–60.36%, and nitrogen fixation from nodules accounted for 39.64–50.01%, indicating that nitrogen in the roots with different phosphorus levels from the initial flowering stage to the initial pod's stage was the main nitrogen supply for fertilizers. In the R5 period, when the phosphorus supply level increased from 1 mg/L to 51 mg/L, nitrogen from fertilizers accounted for 26.14–36.04%, and nitrogen fixation from nodules accounted for 63.96–73.86%. In the R8 period, when the phosphorus supply level increased from 1 mg/L to 51 mg/L, the nitrogen of the roots that came from fertilizer nitrogen accounted for 52.41–68.68%, and the nitrogen fixation from nodules accounted for 29.26–47.59%. The nitrogen source from nodules showed that from the R1-R5 stages, nitrogen from self-fixation accounted for 73.95–91.63%, and nitrogen from fertilizer accounted for only 8.37–26.04%.

**Table 4.** The proportion of nitrogen sources in different parts of soybean under different phosphorus supply levels (%).

| Stages | Treatments | Shoot | | Root | | Nodule | |
|---|---|---|---|---|---|---|---|
| | | Nitrogen Derived from the Fertilizer | Nitrogen Derived from Atmosphere | Nitrogen Derived from the Fertilizer | Nitrogen Derived from Atmosphere | Nitrogen Derived from the Fertilizer | Nitrogen Derived from Atmosphere |
| R1 | P1 | 63.58 ± 0.14 b | 36.42 ± 0.14 e | 68.11 ± 0.16 b | 31.89 ± 0.16 e | 25.21 ± 0.16 a | 74.79 ± 0.09 c |
| | P11 | 65.52 ± 0.25 a | 34.48 ± 0.25 f | 72.47 ± 0.06 a | 27.53 ± 0.06 f | 26.04 ± 1.25 a | 73.95 ± 1.25 c |
| | P21 | 56.90 ± 0.20 c | 43.10 ± 0.20 d | 64.88 ± 0.03 c | 35.12 ± 0.03 d | 19.89 ± 0.88 b | 80.11 ± 0.88 b |
| | P31 | 53.58 ± 0.02 d | 46.42 ± 0.02 c | 64.03 ± 0.19 d | 35.97 ± 0.19 c | 18.94 ± 2.97 b | 81.06 ± 2.97 b |
| | P41 | 39.57 ± 0.15 f | 60.43 ± 0.15 a | 53.01 ± 0.13 f | 46.99 ± 0.13 a | 13.84 ± 0.69 d | 86.16 ± 0.69 a |
| | P51 | 42.67 ± 0.10 e | 57.33 ± 0.10 b | 53.50 ± 0.02 e | 46.50 ± 0.02 b | 16.70 ± 1.09 bc | 83.30 ± 1.09 ab |
| R3 | P1 | 50.80 ± 0.30 c | 49.20 ± 0.30c | 56.78 ± 0.07 c | 43.22 ± 0.07 c | 16.75 ± 0.38 abc | 83.25 ± 0.38 abc |
| | P11 | 54.21 ± 0.38 a | 45.79 ± 0.38 e | 60.36 ± 0.06 a | 39.64 ± 0.06 e | 18.43 ± 0.27 a | 81.57 ± 0.27 c |
| | P21 | 51.51 ± 0.16 b | 48.49 ± 0.16 d | 59.10 ± 0.16 b | 40.90 ± 0.16 d | 17.58 ± 0.40 ab | 82.42 ± 0.40 bc |
| | P31 | 45.72 ± 0.05 d | 54.28 ± 0.05 b | 55.74 ± 0.27 d | 44.26 ± 0.27 b | 14.67 ± 0.84 bc | 85.33 ± 0.84 ab |
| | P41 | 45.85 ± 0.07 d | 54.15 ± 0.07 b | 55.85 ± 0.35 d | 44.15 ± 0.35 b | 15.53 ± 1.24 abc | 84.47 ± 1.24 abc |
| | P51 | 37.91 ± 0.04 e | 62.09 ± 0.04 a | 49.99 ± 0.08 e | 50.01 ± 0.08 a | 13.53 ± 1.94 c | 86.47 ± 1.94 a |
| R5 | P1 | 34.03 ± 0.12 c | 65.97 ± 0.12 c | 34.00 ± 0.12 b | 66.00 ± 0.10 d | 9.89 ± 0.58 ab | 90.11 ± 0.59 ab |
| | P11 | 36.50 ± 0.22 a | 63.50 ± 0.22 e | 36.04 ± 0.69 a | 63.96 ± 0.69 e | 10.78 ± 0.45 a | 89.22 ± 0.45 b |
| | P21 | 34.88 ± 0.07 b | 65.12 ± 0.07 d | 32.15 ± 0.17 c | 67.85 ± 0.17 c | 10.38 ± 0.26 ab | 89.62 ± 0.26 ab |
| | P31 | 29.30 ± 0.19 d | 70.70 ± 0.19 b | 29.91 ± 0.22 d | 70.09 ± 0.22 b | 8.37 ± 0.28 b | 91.63 ± 0.28 a |
| | P41 | 27.42 ± 0.15 e | 72.58 ± 0.15 a | 27.10 ± 0.14 e | 72.89 ± 0.80 a | 9.33 ± 1.83 ab | 90.67 ± 1.83 ab |
| | P51 | 27.18 ± 0.08 e | 72.81 ± 0.08 a | 26.14 ± 0.26 e | 73.86 ± 0.26 a | 9.57 ± 1.16 ab | 90.43 ± 1.16 ab |
| R8 | P1 | 68.22 ± 1.30 a | 31.78 ± 1.30 c | 68.68 ± 0.67 a | 31.32 ± 0.67 b | - | - |
| | P11 | 72.35 ± 1.38 a | 27.65 ± 1.38 c | 70.74 ± 1.33 a | 29.26 ± 1.33 b | - | - |
| | P21 | 61.77 ± 1.57 b | 38.23 ± 1.57 b | 58.73 ± 0.36 b | 41.27 ± 0.36 a | - | - |
| | P31 | 52.68 ± 3.43 c | 47.32 ± 3.43 a | 52.41 ± 3.00 b | 47.59 ± 3.00 a | - | - |
| | P41 | 59.69 ± 1.62 b | 40.31 ± 1.62 b | 54.95 ± 5.70 b | 45.05 ± 5.70 a | - | - |
| | P51 | 58.28 ± 0.42 bc | 41.71 ± 0.42 ab | 53.20 ± 1.71 b | 46.80 ± 1.71 a | - | - |

Different lowercase letters indicate the differences between treatments at a significance level of 5%.

### 3.2.3. Nitrogen Accumulation and Nodule Nitrogen Fixation in Different Parts of Soybean under Different Phosphorus Supply Levels

According to the dry matter accumulation in each part of the soybean and the nitrogen content per gram of dry matter, the nitrogen accumulation in each part can be identified, and the nitrogen accumulation in each part multiplied by the nitrogen fixation rate of the nodule (Table 4) can be obtained. Table 5 shows that the nitrogen accumulation in all parts of soybean increased linearly with the growth process, and the nitrogen fixation in nodules increased from the R1 stage to the R5 stage and decreased from the R5 stage to the R8 stage (except nodules). The different phosphorus supply treatments generally showed a significant increase with increasing phosphorus supply levels. Table 5 shows that the nitrogen content of the shoots and roots reached a maximum when the phosphorus supply level increased from 1 mg/L to 31 mg/L at the R1 stage, and then the phosphorus supply level showed a decreasing trend. In the R3 stage, when the phosphorus supply level increased to 21 mg/L, there was no significant upward trend in increasing the phosphorus supply level. The R5 and R8 stages increased with the increase in the phosphorus supply level, but when the phosphorus supply level increased to 31–41 mg/L, there was no significant upward trend in increasing the phosphorus supply level. The initial flowering stage was 31 mg/L, the initial pod stage was 21 mg/L, and the seed-filling stage and maturity stage were both 31 mg/L–41 mg/L. However, the nitrogen accumulation in nodules increased with increasing phosphorus supply levels from the R1 stage to the R5 stage. When the phosphorus supply level increased by 41 mg/L, there was no significant increasing trend when the phosphorus supply level was increased. It can also be concluded from Table 5 that the nitrogen fixation amount of the shoot increases to a maximum of 41 mg/L in the R1 stage, and there is a downward trend when the phosphorus supply level is increased again. From the R3 stage to the R5 stage, it increased with increasing phosphorus supply level, and there was no significant increase between 31 mg/L and 51 mg/L in the R3 stage. There was no significant increase between 41 mg/L and 51 mg/L in the R5 stage. In the R8 stage, when the phosphorus supply level increased to 31 mg/L, the maximum level increased, and the phosphorus supply level showed a decreasing trend. This indicated that it was not the case that the higher the concentration of phosphorus supply was, the better the accumulation of nitrogen fixation in soybean nodules. The phosphorus supply level of 31 mg/L–41 mg/L was the most suitable phosphorus supply level to promote the efficient utilization of nitrogen fixation by nodules in the shoots of soybean. The nitrogen fixation amount by nodules in soybean was the largest in the 31 mg/L treatments in the R1, R3, and R8 stages, while it was 41 mg/L in the R5 stage. The nitrogen fixation of nodules showed that from the R1–R5 stages, it increased with the progression of the growth period and increased with the increase in the phosphorus supply level in the same period, but there was no significant difference when the phosphorus supply level reached 31 mg/L in the R1 stage and no significant increase when the phosphorus supply level reached 41 mg/L in the R3 and R5 stages.

**Table 5.** Nitrogen accumulation and nodule nitrogen fixation in different parts of soybean under different phosphorus supply levels (mg).

| Stages | Treatments | Shoot | | Root | | Nodule | |
| | | Nitrogen Accumulation | Nodule Nitrogen Fixation | Nitrogen Accumulation | Nodule Nitrogen Fixation | Nitrogen Accumulation | Nodule Nitrogen Fixation |
|---|---|---|---|---|---|---|---|
| R1 | P1 | 115.89 ± 4.40 c | 42.20 ± 1.46 d | 47.57 ± 2.39 a | 15.18 ± 0.80 b | 2.88 ± 0.23 d | 2.15 ± 0.17 c |
| | P11 | 130.85 ± 9.18 c | 45.16 ± 3.46 d | 30.62 ± 0.81 c | 8.43 ± 0.23 d | 3.47 ± 0.30 d | 2.57 ± 0.26 c |
| | P21 | 160.71 ± 5.04 b | 69.24 ± 1.87 c | 36.78 ± 1.98 b | 12.92 ± 0.71 c | 6.29 ± 0.26 c | 5.03 ± 0.16 b |
| | P31 | 228.20 ± 12.27 a | 105.93 ± 5.65 b | 48.60 ± 0.31 a | 17.48 ± 0.14 a | 10.19 ± 0.34 b | 8.28 ± 0.55 a |
| | P41 | 224.62 ± 6.12 a | 135.74 ± 3.86 a | 36.55 ± 1.63 b | 17.17 ± 0.79 a | 10.50 ± 0.66 ab | 9.05 ± 0.61 a |
| | P51 | 209.76 ± 15.56 a | 120.28 ± 9.10 b | 31.38 ± 0.69 c | 14.59 ± 0.32 bc | 11.58 ± 0.54 a | 9.65 ± 0.56 a |

**Table 5.** *Cont.*

| Stages | Treatments | Shoot | | Root | | Nodule | |
| | | Nitrogen Accumulation | Nodule Nitrogen Fixation | Nitrogen Accumulation | Nodule Nitrogen Fixation | Nitrogen Accumulation | Nodule Nitrogen Fixation |
|---|---|---|---|---|---|---|---|
| R3 | P1 | 215.25 ± 5.93 b | 109.03 ± 3.14 d | 70.37 ± 3.62 b | 30.41 ± 1.52 c | 4.96 ± 0.33 d | 4.13 ± 0.26 d |
| | P11 | 342.50 ± 19.47 b | 156.84 ± 9.11 cd | 70.60 ± 1.10 b | 27.98 ± 0.46 c | 7.56 ± 0.30 d | 6.17 ± 0.23 d |
| | P21 | 523.62 ± 51.90 a | 253.83 ± 24.76 bc | 84.30 ± 3.28 a | 34.48 ± 1.42 b | 23.75 ± 0.69 c | 19.57 ± 0.53 c |
| | P31 | 543.99 ± 91.92 a | 295.34 ± 50.67 ab | 89.23 ± 3.65 a | 39.48 ± 1.48 a | 30.06 ± 6.79 bc | 25.68 ± 5.94 bc |
| | P41 | 580.62 ± 31.39 a | 314.44 ± 17.18 ab | 85.69 ± 2.36 a | 37.82 ± 0.76 ab | 37.60 ± 2.11 ab | 31.72 ± 1.37 ab |
| | P51 | 620.71 ± 86.00 a | 385.32 ± 53.19 a | 71.36 ± 0.66 b | 35.68 ± 0.33 b | 42.92 ± 2.61 a | 37.01 ± 1.41 a |
| R5 | P1 | 352.59 ± 17.24 d | 232.64 ± 11.78 e | 90.12 ± 8.83 b | 59.48 ± 5.84 c | 6.65 ± 0.52 c | 5.97 ± 0.43 c |
| | P11 | 709.27 ± 28.87 c | 450.28 ± 17.11 d | 98.52 ± 6.85 ab | 63.07 ± 4.83 c | 12.45 ± 2.47 c | 11.13 ± 2.27 c |
| | P21 | 919.26 ± 91.54 b | 598.51 ± 59.18 c | 105.06 ± 9.12 ab | 71.27 ± 6.06 bc | 37.87 ± 1.75 b | 33.93 ± 1.51 b |
| | P31 | 1046.43 ± 9.18 b | 739.81 ± 6.68 b | 119.72 ± 7.01 a | 83.94 ± 5.10 ab | 48.97 ± 7.83 b | 44.83 ± 7.08 b |
| | P41 | 1412.70 ± 68.32 a | 1025.37 ± 49.56 a | 123.97 ± 6.26 a | 90.37 ± 4.23a | 69.74 ± 8.22 a | 63.16 ± 7.09 a |
| | P51 | 1425.14 ± 78.06 a | 1037.73 ± 56.37 a | 102.40 ± 7.08 ab | 75.65 ± 5.34 abc | 77.19 ± 5.54 a | 69.76 ± 4.70 a |
| R8 | P1 | 365.04 ± 6.40 e | 115.86 ± 2.70 d | 80.63 ± 1.64 c | 25.27 ± 1.03 c | - | - |
| | P11 | 912.96 ± 54.74 d | 253.77 ± 26.37 d | 94.43 ± 2.10 bc | 27.64 ± 1.56 c | - | - |
| | P21 | 1388.97 ± 63.41 c | 532.34 ± 42.69 c | 113.47 ± 7.86 abc | 46.77 ± 2.86 b | - | - |
| | P31 | 1949.42 ± 160.16 ab | 927.12 ± 123.85 a | 145.47 ± 7.13 a | 69.06 ± 4.48 a | - | - |
| | P41 | 2022.57 ± 70.08 a | 815.01 ± 41.46 ab | 125.81 ± 16.35 ab | 55.20 ± 5.60 ab | - | - |
| | P51 | 1711.01 ± 73.11 b | 713.99 ± 33.22 b | 106.95 ± 23.99 ab | 49.28 ± 9.24 b | - | - |

Different lowercase letters indicate the differences between treatments at a significance level of 5%.

### 3.3. Comparisons between ARA, RNNF%, and Naccumulation of Nodules

Figure 1 shows the changing trend of the soybean ARA nitrogen fixation rate and nitrogen fixation amount of soybean nodules under different phosphorus supply levels at different growth stages. In terms of the growth process, the ARA, the nitrogen fixation rate of the whole nodule, and the nitrogen fixation amount of the whole nodule of soybean in the R1, R5, and R8 stages increased first and then decreased with the increase in the phosphorus supply level, peaked at 31 mg/L–41 mg/L phosphorus supply level.

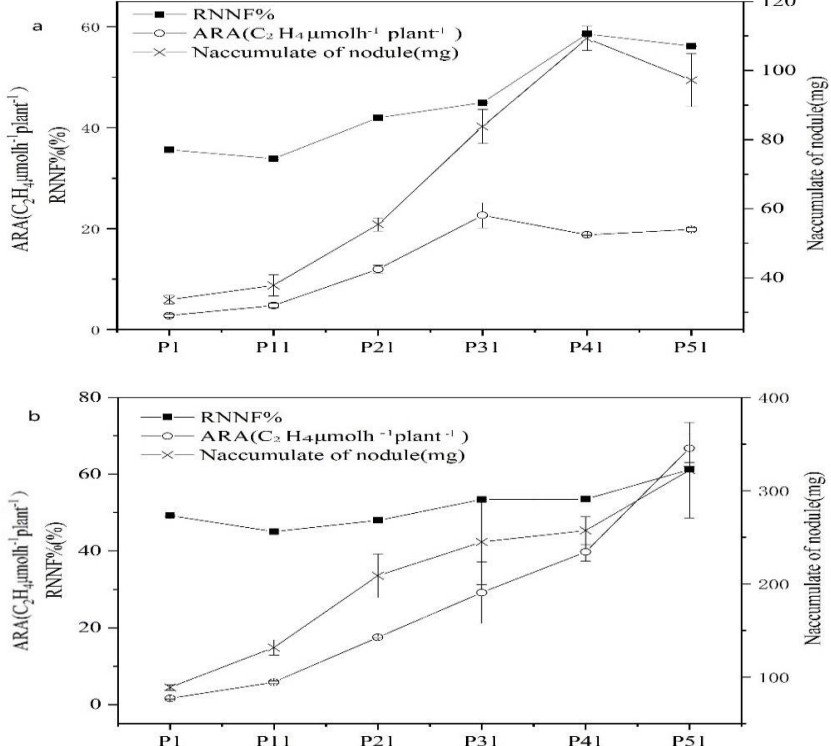

**Figure 1.** *Cont.*

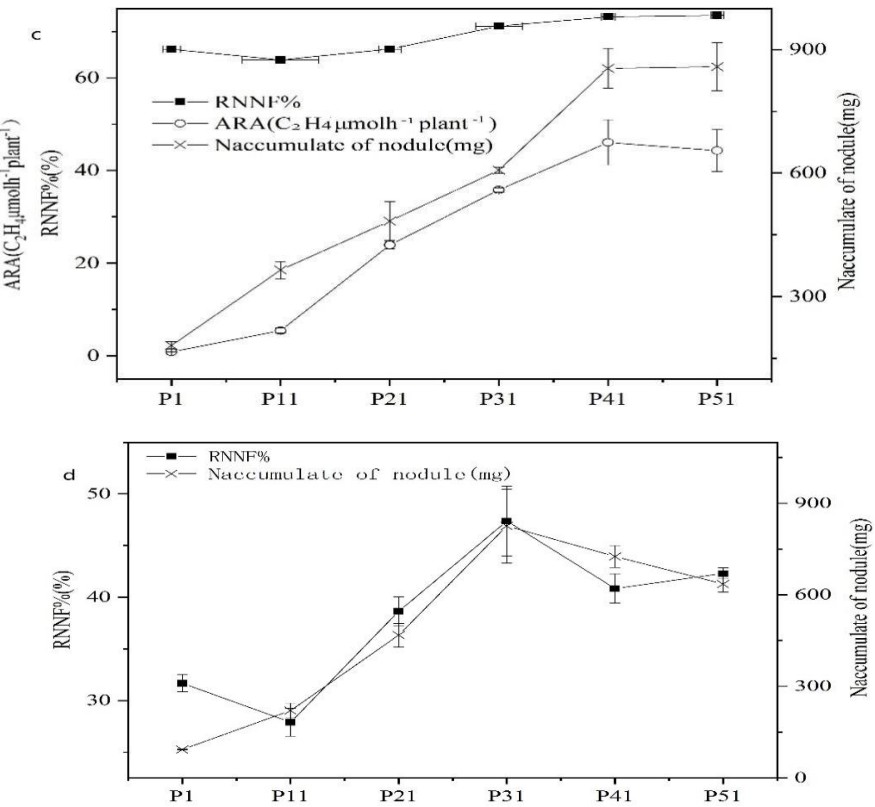

**Figure 1.** Variation trends of ARA, root nodule nitrogen fixation rate, and root nodule nitrogen fixation in soybean under different phosphorus supply levels. (**a**): initial flowering period; (**b**): initial pod stage; (**c**): filling period; (**d**): mature period.

*3.4. Correlations Analysis between Different Phosphorus Supply Levels and Soybean ARA, Nitrogen Fixation Rate of Nodules, and Nitrogen Fixation of Nodules*

Table 6 shows the correlation between the phosphorus supply level at different growth stages and ARA, whole nodule nitrogen fixation rate, and whole nodule nitrogen fixation amount. The correlation performance changes in each growth period were consistent, which all showed that the level of phosphorus supply was significantly positively correlated with ARA, the nitrogen fixation rate of the whole nodule, and the nitrogen fixation rate of the whole nodule ($R^2 \geq 0.803$). There was a very significant positive correlation ($R^2 \geq 0.791$) in the nitrogen fixation of the nodules of the whole plant, while the nitrogen fixation rate of the whole plant nodules and the nitrogen fixation of the whole plant nodules had a very significant positive correlation ($R^2 \geq 0.847$) in the R1 to R5 stages and were significantly positively correlated with the nitrogen fixation amount of whole nodules in the R8 stage ($R^2 = 0.725$).

**Table 6.** Correlation of different phosphorus supply levels, ARA, nitrogen fixation rate of nodules, and nitrogen fixation of nodules.

| Stages | | Phosphorus Supply Level | ARA | Whole Nodule Nitrogen Fixation Rate |
|---|---|---|---|---|
| R1 | nodule nitrogen fixation amount | 0.913 ** | 0.876 ** | 0.966 ** |
| | whole nodule nitrogen fixation rate | 0.902 ** | 0.791 ** | - |
| | ARA | 0.978 ** | - | - |
| R3 | nodule nitrogen fixation amount | 0.987 ** | 0.930 ** | 0.847 ** |
| | whole nodule nitrogen fixation rate | 0.883 ** | 0.977 ** | - |
| | ARA | 0.966 ** | - | - |
| R5 | nodule nitrogen fixation amount | 0.992 ** | 0.981 ** | 0.875 ** |
| | whole nodule nitrogen fixation rate | 0.847 ** | 0.837 ** | - |
| | ARA | 0.971 ** | - | - |
| R8 | nodule nitrogen fixation amount | 0.952 ** | - | 0.725 * |
| | whole nodule nitrogen fixation rate | 0.803 ** | - | - |

* denote a significant difference at the 5% level; ** denote a significant difference at the 1% level.

## 4. Discussion

Most studies have shown that phosphorus can promote nodule growth and nitrogenase activity in leguminous crops, showing that with the increase in phosphorus supply, the number and dry weight of nodules and the activity of nodule nitrogenase in leguminous crops are significantly increased [26–33]. In this experiment, it was found that 41 mg/L was the optimum phosphorus supply level for soybean nodule growth, and low phosphorus concentration (1 mg/L–31 mg/L) promoted the phenomenon of "survival of the fittest" in soybean nodules. This is generally consistent with the results of Miao et al. [34] and Jemo et al. [29], who found that soybean nodule dry weight, nodule number, and nodule nitrogenase all decreased with the decrease in phosphorus supply level, and Yao et al. [23] found that phosphorus levels that were too high or too low inhibited the nitrogen fixation of soybean nodules; this was similar to the conclusion of Tsvetkova et al. [35] and Magadlela et al. [7] that low phosphorus concentration inhibited the nitrogen fixation of nodules by inhibiting the growth and phosphorus absorption of soybean plants, but differed from the conclusion of Almeida et al. [13] that low phosphorus concentration can stimulate the growth of soybean nodules and inhibit nitrogen fixation by nodules. This may be due to the different levels of low-phosphorus stress or the change in the direction of phosphorus transport caused by low-phosphorus stress. This indicates that different phosphorus supply levels will significantly affect the nitrogen fixation of soybean nodules. In addition, this experiment also found that the sources of nitrogen in different parts of soybean showed different trends with the growth period and phosphorus supply concentration. Among them, from the initial flowering stage to the filling stage, the main body of nitrogen supply in the low phosphorus treatment (1 mg/L–31 mg/L) will gradually change from fertilizer nitrogen to nodule nitrogen fixation, while the main body of nitrogen supply in the high phosphorus treatment (41 mg/L–51 mg/L) will always be nodule nitrogen fixation and will be transformed into fertilizer nitrogen at the mature stage. The main nitrogen supply of soybean roots at different levels of phosphorus supply from the initial flowering to the initial pods and maturity stage was fertilizer nitrogen, and the main nitrogen supply at the seed filling stage was nodule nitrogen fixation (as shown in Figure 2). This is different from the findings of Raji et al. [36], Schulze et al. [37], and Cavard et al. [38] that under soil culture conditions, the phosphorus supply level had no significant effect on the nitrogen fixation rate of leguminous nodules and is also different from the research conclusion drawn by Magadlela et al. [28] that 40–50% of the nitrogen nutrition absorbed by plants comes from symbiotic nitrogen fixation. This may be because the soil itself contains a certain amount of phosphorus under soil culture conditions, which affects the phosphorus supply gradient of the experimental setup. Then increased, the level of phosphorus indicators

with a decreasing trend, while the indicators in the R3 stage increased with the increase in phosphorus supply level. This shows that the phosphorus supply level required for nitrogen fixation of nodules at different growth stages is different. The phosphorus supply level of 31 mg/L–41 mg/L in the R1 and R5-R8 stages is the optimal phosphorus supply level to promote nitrogen fixation of soybean nodules, while the R3 stage is the period when the nodule grows vigorously. The phosphorus supply required for nitrogen fixation by nodules was significantly higher than that in other growth periods. From the changing trend of each index, the nitrogen fixation of ARA and the nitrogen fixation of the whole plant showed the same change rule in each growth period. This shows that both ARA and $^{15}$N tracer methods determined by the acetylene reduction method are effective methods for determining nitrogen fixation in nodules. At the 1 mg/L–11 mg/L phosphorus supply level, ARA and the nitrogen fixation rate of the whole root nodule showed opposite trends. This result indicated that the low phosphorus supply would stimulate nodule nitrogen fixation, resulting in the opposite trend of the nitrogen fixation rate of ARA and whole nodules under the phosphorus supply level of 1 mg/L–11 mg/L. In addition, through the acetylene reduction method in this experiment, it was concluded that the nitrogen fixation of soybean nodules in different growth stages has different requirements for phosphorus supply. The initial flowering stage and seed-filling stage were 31 mg/L–41 mg/L, and the initial pod stage was 51 mg/L. The $^{15}$N tracer method showed that the optimal phosphorus supply concentration for promoting nitrogen accumulation and efficient utilization of nitrogen fixation in soybean nodules at each growth stage was 31 mg/L–41 mg/L. The optimum phosphorus concentrations for nitrogen fixation of soybean nodules at different growth stages measured by the two methods are relatively close, which indicates that the acetylene reduction method and the $^{15}$N tracer method are both effective methods for determining the nitrogen fixation capacity of soybean nodules.

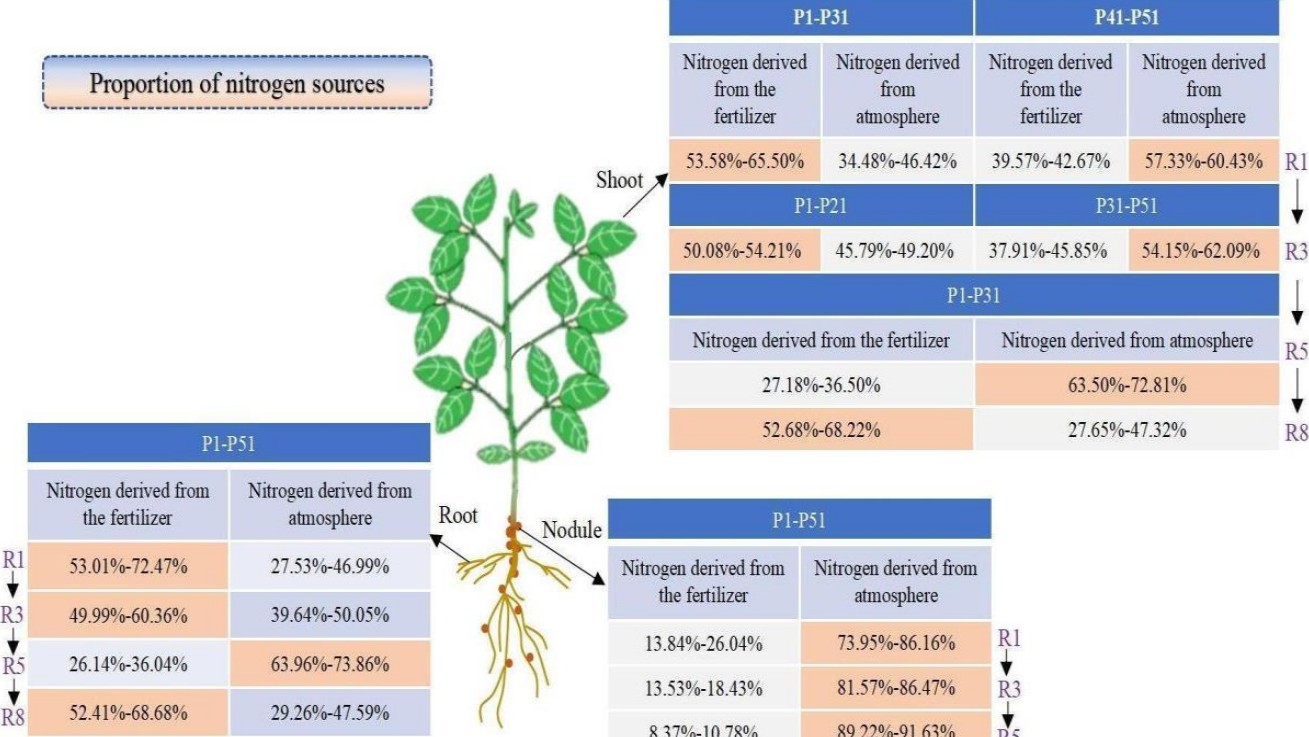

**Figure 2.** The proportion of nitrogen sources in soybean organs in different growth stages under different phosphorus supply levels.

## 5. Conclusions

1. The optimum phosphorus supply level for soybean nodule growth is 41 mg/L, and the acetylene reduction method also shows that the nitrogen fixation of soybean nodules at different growth stages has different requirements for phosphorus supply level, which is 31 mg/L–41 mg/L at the initial flowering and seed filling stages and 51 mg/L at the initial pod stage.

2. The nitrogen source in different parts of soybean showed different trends with different growth periods and phosphorus supply concentrations. Among them, from the initial flowering stage to the seed filling stage, the main body of the nitrogen supply of soybean shoots in the low phosphorus treatment (1 mg/L–31 mg/L) will gradually change from fertilizer nitrogen to nodule nitrogen fixation, while the main body of the nitrogen supply of soybean shoots in the high phosphorus treatment (41 mg/L–51 mg/L) will always be nodule nitrogen fixation and will be transformed into fertilizer nitrogen at the mature stage. The main nitrogen supply of soybean roots at different levels of phosphorus supply from the initial flowering to the initial pods and maturity stage was fertilizer nitrogen, and the main nitrogen supply at the seed filling stage was nodule nitrogen fixation. The nitrogen supply to the main body of soybean nodules is always nodule nitrogen fixation.

3. Different phosphorus supply levels can significantly affect the nitrogen fixation of soybean nodules ($R^2 \geq 0.803$), and both the acetylene reduction method and the $^{15}N$ tracer method can be used to determine the nitrogen fixation ability of soybean nodules.

**Supplementary Materials:** The following supporting information can be downloaded at: https://www.mdpi.com/article/10.3390/agronomy12112802/s1, Table S1: The elements of P or K in different P level treatments; Table S2: Concentrations of elements in nutrient medium of the sand culture; Table S3: Amount of nutrient solution for soybean at different growth stages.

**Author Contributions:** Writing review and editing, H.L. and Z.Z., funding acquisition and investigation, L.W., A.Y. and D.L. All authors have read and agreed to the published version of the manuscript.

**Funding:** We are grateful for the "Young Talents" Project of Northeast Agricultural University, award number: 19QC16, and the Postdoctoral Scientific Research Development Fund of Heilongjiang Province, award number: LBH-Z17033.

**Institutional Review Board Statement:** The study was conducted in accordance with the Declaration of Helsinki, and approved by the Institutional Review Board (or Ethics Committee) of NAME OF INSTITUTE.

**Informed Consent Statement:** Informed consent was obtained from all subjects involved in the study.

**Data Availability Statement:** Not Applicable.

**Conflicts of Interest:** The authors declare no conflict of interest.

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
