# Peer review of "Effect of Phosphorus Supply Levels on Nodule Nitrogen Fixation and Nitrogen Accumulation in Soybean (Glycine max L.)"

_agronomy, doi:10.3390/agronomy12112802_

Round 1
Reviewer 1 Report
Line 17: "nitrogen content" of what?
Line 18: exceed space in"1. 41"
Lines 17 and 18: "(R1, 17 R3, R5, and R8)" please defined what it means in the abstract.
Line 19: this phrase "low phosphorus stress" is not accurate Because it is not correct to describe stress as low, either there is stress or there is no stress. you can replace it with "low phosphorus concentration". This should be taken into account in the rest of the manuscript.
Lines 54 and 55: what is the difference or the addition you made compared to what you wort in lines 59 and 60? I think it is repeated.
Lines 93 and 93: the proposal of dividing the pots with polycarbonate plastic plates is not clear. please clarify your propose.
Line 96: it is not reasonable that a plastic pot 30 cm in diameter and 28 cm in height is filled with "20 kg" of sand. please recheck the weight.
Line 101: is it one seedling per each half of the pot? please defined the number of plants or pots per each treatment
Lines 104 and 105: the abbreviation of VC is not in harmony with the term is in parentheses "(the unfolded cotyledon stage)" I think it should be UC or UCS.
Line 103-115: it is better if you make a table containing the V and R stages and the nutrient supply for each.
Line 115-117: you should mention the symbiotic bacteria you used which was Rhizobium. also, move these lines to above in the paragraph before you mention the nutrient supply.
Line 119: what is the meaning of "R1, R3, R5, and R8" please defined them clearly with mention the period of each stage in days.
Line 120: add AM after "8:00 and 10:00"
Line 122: mention the reference of nitrogenase activity after you mention it. please rearrange it.
Line 126-128: I did not hear before about this method for measuring the dry matter of leaves through collecting the falling leaves. if so please mention the reference of this method. or you have another point of view, please clarify it in this part of the materials and methods. After all, this measurement did not mention in the results
line 141: you mentioned the measurement of "The nitrogenase activity per unit nodule weight" but the equation abbreviation was SNR; whereas, you write its abbreviation of SNA in the rest of the manuscript, which one is right? However, the abbreviation should be in harmony with its term by using the initials of the term. for example, the term "The nitrogenase activity per unit nodule weight" may be NAN. The same case is in ARA, which is supposed to express the term "single plant nitrogenase activity". Or you can modify the terms as mentioned in the reference method.
Line156: you did not mention the statistical design you used e.g., CRD, RCBD... etc. in split of factorial arrangement, it is supposed to put the rhizobium with two treatments (inoculated and uninoculated) and under each one put the phosphorous levels. In addition to mentioning the number of replication and the number of plants or pots per treatment.
Line 174-178, 180-182, 197-199 ...etc.: when the manuscript is divided into two titles as results - discussion, you should not comment on the results in anyhow. please delete your comments from the results and write them under the discussion title. Also, this should be taken into account in the rest of the manuscript.
Line 178: "dry weight of soybean" of what... plant or nodules, please clarify.
Line 204: "initial flowering and germination stages" it did not mention in the materials and methods.
In general, if you adopt a specific term or a specific abbreviation in the materials and methods, you should use it in the rest of the manuscript so as not to confuse the reader.
Line 277: "initial flowering stage to the sprouting stage" please mention its abbreviation.
Line 321: "the seed filling stage" it did not mention in the materials and methods.
Line 330-331: the results of the figures did not match with the title " correlation ...." that related to the results in Table 6.
Lines 342-356: move it to the discussion
Lines 410-414: move your comment to the discussion.
Line 433: "The nitrogen supply to the main body of soybean nodules is always nodule nitrogen fixation" what is your point of view, it is vague.
In general, please use short and clear sentences in the manuscript as you can. Also, it is preferred to shorten the conclusion to be more clear and more precise. Moreover, there is no need to use the term “survival of the fittest” because it is not suitable for the study issue and it looks strange on the context.
Author Response
Dear Editors and Reviewers:
We would like to thank you for the constructive and helpful comments we received, which helped us to improve the quality of our manuscript. In this new version, we have taken into account all the comments raised by the reviewers, as described in the list of detailed responses provided. We have revised the author, introduction,keywords, text, author contributions, conflicts of interest in the manuscript as required, and added the supplementary materials to make the manuscript more completed.
Point 1: Line 17: "nitrogen content" of what?
Response 1: Thanks for your advice, we have added nitrogen content in line 16.
Point 2: Line 18: exceed space in"1. 41"
Response 2: Thanks for your suggestion, we have deleted the exceed space part ,20-22 line.
Point 3: Lines 17 and 18: "(R1, 17 R3, R5, and R8)" please defined what it means in the abstract.
Response 3: Thanks for your suggestion, we have defined R1, R3, R5 and R8 in line 18-19.
Point 4: Line 19: this phrase "low phosphorus stress" is not accurate Because it is not correct to describe stress as low, either there is stress or there is no stress. you can replace it with "low phosphorus concentration". This should be taken into account in the rest of the manuscript.
Response 4: Thank you very much for your suggestion, we have replaced “low phosphorus stress” with “low phosphorus concentration”, 54,58,60 line .
Point 5: Lines 54 and 55: what is the difference or the addition you made compared to what you wort in lines 59 and 60? I think it is repeated.
Response 5: Thank you for your comments, the latter part is a detailed explanation of the previous part, and we have added characterized by at the beginning of the latter part in 63 lines.
Point 6: Lines 93 and 93: the proposal of dividing the pots with polycarbonate plastic plates is not clear. please clarify your propose.
Response 6: I am very sorry for the incorrect expression of this part, and we have deleted the wrong part, line101-104.
Point 7: Line 96: it is not reasonable that a plastic pot 30 cm in diameter and 28 cm in height is filled with "20 kg" of sand. please recheck the weight.
Response 7: We have rechecked according to your requirements and change river sand to washed sand in 105 lines, because the sand is cleaned before being loaded into the pot, so the weight is heavier.
Point 8: Line 101: is it one seedling per each half of the pot? please defined the number of plants or pots per each treatment
Response 8: Thank you for your suggestion, mentioning 2 seedlings were kept per pot in line 111
Point 9: Lines 104 and 105: the abbreviation of VC is not in harmony with the term is in parentheses "(the unfolded cotyledon stage)" I think it should be UC or UCS.
Response 9: Thank you very much for your suggestion, the abbreviation of VC is correct, quoted from Fehr, W.R. ; caviness, C.E. ; burmood, D.T. ; pennington, J.S. Stage of Development Descriptions for Soybeans, Glycine Max ( L.) Merrill. Crop. Sci. 1971, 11, 929-931.
Point 10: Line 103-115: it is better if you make a table containing the V and R stages and the nutrient supply for each.
Response 10: According to the reviewer 's comments, we have added Tab S3 containing the V and R stages and the nutrient supply for each.
Point 11: Line 115-117: you should mention the symbiotic bacteria you used which was Rhizobium. also, move these lines to above in the paragraph before you mention the nutrient supply.
Response 11: According to the reviewer 's comments, we have moved this part to the beginning of the paragraph, 112-115 lines.
Point 12: Line 119: what is the meaning of "R1, R3, R5, and R8" please defined them clearly with mention the period of each stage in days.
Response 12: Thank you very much for your suggestion, we have defined meaning of ' R1, R3, R5, and R8 ' and added the period of each stage in days in Line 132-134.
Point 13: Line 120: add AM after "8:00 and 10:00"
Response 13: According to the reviewer 's comments, we have added AM after 8 : 00 and 10 : 00, 135 line
Point 14: Line 122: mention the reference of nitrogenase activity after you mention it. please rearrange it.
Response 14: Thanks to expert’s advice, we have added references in 139 lines
Point 15: Line 126-128: I did not hear before about this method for measuring the dry matter of leaves through collecting the falling leaves. if so please mention the reference of this method. or you have another point of view, please clarify it in this part of the materials and methods. After all, this measurement did not mention in the results
Response 15: Thanks to the reviewer 's comments, we have added 143 lines of explanation to the material method according to expert opinion.
Point 16: line 141: you mentioned the measurement of "The nitrogenase activity per unit nodule weight" but the equation abbreviation was SNR; whereas, you write its abbreviation of SNA in the rest of the manuscript, which one is right? However, the abbreviation should be in harmony with its term by using the initials of the term. for example, the term "The nitrogenase activity per unit nodule weight" may be NAN. The same case is in ARA, which is supposed to express the term "single plant nitrogenase activity". Or you can modify the terms as mentioned in the reference method.
Response 16: Thanks to the reviewers, we have modified the abbreviations of SNA and ANA according to your comments,in lines 164 and 170.
Point 17: Line156: you did not mention the statistical design you used e.g., CRD, RCBD... etc. in split of factorial arrangement, it is supposed to put the rhizobium with two treatments (inoculated and uninoculated) and under each one put the phosphorous levels. In addition to mentioning the number of replication and the number of plants or pots per treatment.
Response 17: Thank you for your suggestion, we mentioned the number of replicates in line 132, and mentioned the inoculation in lines 112-115. In the 2.3 Experimental design section, the experimental treatment of this experiment is phosphorus concentration treatment.
Point 18: Line 174-178, 180-182, 197-199 ...etc.: when the manuscript is divided into two titles as results - discussion, you should not comment on the results in anyhow. please delete your comments from the results and write them under the discussion title. Also, this should be taken into account in the rest of the manuscript.
Response 18: According to the reviewer 's opinion, the explanation part of each paragraph of the result has been deleted, Lines 196-199, 203-202, 292-296, 300-303 305-307, 311-312,323-325, 327-329, 459-461.
Point 19: Line 178: "dry weight of soybean" of what... plant or nodules, please clarify.
Response 19: We have supplemented the parts of dry weight according to the requirements of the reviewers, in line 200.
Point 20: Line 204: "initial flowering and germination stages" it did not mention in the materials and methods. In general, if you adopt a specific term or a specific abbreviation in the materials and methods, you should use it in the rest of the manuscript so as not to confuse the reader.
Response 20: Thank you for your comments, this part has been deleted as the explanation part of the result analysis.
Point 21: Line 277: "initial flowering stage to the sprouting stage" please mention its abbreviation.
Response 21: Thank you for your comments, this part has been deleted as the explanation part of the result analysis, Line 327-329.
Point 22: Line 321: "the seed filling stage" it did not mention in the materials and methods.
Response 22: Thank you for your comments. This part has been deleted as the explanation part of the result analysis
Point 23: Line 330-331: the results of the figures did not match with the title " correlation ...." that related to the results in Table 6.
Response 23: According to the reviewer 's opinion, we have added the title in line 413.
Point 24: Lines 342-356: move it to the discussion
Response 24: We have moved this section to the discussion in lines 496-509.
Point 25: Lines 410-414: move your comment to the discussion.
Response 25: Thanks to the expert's opinion, we have comment on the manuscript on lines 482-485
Point 26: Line 433: "The nitrogen supply to the main body of soybean nodules is always nodule nitrogen fixation" what is your point of view, it is vague.
Response 26: Thank the reviewers for their comments, the data support and analysis for this conclusion are in line 325-327.
Point 27: In general, please use short and clear sentences in the manuscript as you can. Also, it is preferred to shorten the conclusion to be more clear and more precise. Moreover, there is no need to use the term “survival of the fittest” because it is not suitable for the study issue and it looks strange on the context.
Response 27: This section has been deleted according to expert opinion, 523-525 lines
Reviewer 2 Report
The subject of the manuscript corresponds to the subject of the journal and it can be considered for publication. However, the text of the manuscript should be substantially improved. Below I will give some specific comments and questions.
L35: Please add a short conclusion (1-2 sentences) about the scientific significance of this study in the abstract.
L71: Please provide your hypothesis
L85: Why were these phosphorus concentrations chosen when planning the experiment?
L90: I believe that this subsection should be combined with the previous one, because it is also a description of the design of the experiment
L96: Is it really possible to put 20 kg of sand in such a small pot? What was the density of the soil in the experimental pots?
L118: measuremenT (typo)
L161 and 206: I believe these subheadings can be omitted
L163: When describing the results, use only the past tenses (not the presents), because. you describe phenomena, not patterns!
L 187, 208 and 284: Text first, then tables
In general, the description of the results can be slightly reduced, but at the same time it is more clear to indicate which effects were statistically confirmed and which were not.
L373: The Discussion is too short. Please discuss in more detail the possible mechanisms of influence of phosphorus availability on the studied indicators. It is also worth not only pointing out that your results contradict some previously published data, but also trying to explain why this happened.
Please carefully check the text of the manuscript (many typos). If possible, I would advise showing the manuscript to an English-speaking person before resubmitting.
Author Response
Dear Editors and Reviewers:
Thank you for your letter and for the reviewers’ comments concerning our manuscript entitled “Effect of phosphorus supply levels on nodule nitrogen fixation and Nitrogen accumulation in soybean (Glycine max L.)” . Those comments are all valuable and very helpful for revising and improving our paper, as well as the important guiding significance to our researches. We have studied comments carefully and have made correction which we hope meet with approval. Revised portion are marked by using "Track Changes" function in Microsoft Word. We have revised the author, introduction,keywords , text, author contributions, conflicts of interest in the manuscript as required, and added the supplementary materials to make the manuscript more completed.
Point 1:L35: Please add a short conclusion (1-2 sentences) about the scientific significance of this study in the abstract.
Response 1: Thanks to the reviewer 's comments, we have added a short conclusion it as requested in line 37-40.
Point 2:L71: Please provide your hypothesis
Response 2: Thanks to the opinions of reviewers, we have added tab S3 (127 lines) according to expert opinions, and detailed experimental design is provided in manuscript 2.3 (line 93-98).
Point 3:L85: Why were these phosphorus concentrations chosen when planning the experiment?
Response 3: We have added the reference in 96 lines according to the opinions of reviewers.
Point 4:L90: I believe that this subsection should be combined with the previous one, because it is also a description of the design of the experiment
Response 4: Thanks to the reviewer 's comments, 2.4 is indeed the specific treatment of 2.3, and in order to make a clearer distinction between Experimental design and Test treatment, we divide it into two separate sections.
Point 5:L96: Is it really possible to put 20 kg of sand in such a small pot? What was the density of the soil in the experimental pots?
Response 5: Thanks for the opinion of the reviewer, we have changed the river sand to of washed sand (105 lines), because the sand is cleaned before being loaded into the pot, so the weight is heavier.
Point 6:L118: measuremenT(typo)
Response 6: We are very sorry for the spelling error here, we have corrected it (line131).
Point 7:L161 and 206: I believe these subheadings can be omitted
Response 7: We have omitted the subheadings of 3.11 and 3.12 according to the opinion of the reviewer (lines 184 and 209).
Point 8:L163:When describing the results, use only the past tenses (not the presents), because. you describe phenomena, not patterns!
Response 8: We have changed it into the past tenses according to the review expert's opinion (lines 185, 210, 246, 278, 423, 450).
Point 9:L 187, 208 and 284: Text first, then tables
Response 9: According to the opinion of the reviewer, we have placed table 2、table 3 and table 5 after the corresponding text.
Point 10:In general, the description of the results can be slightly reduced, but at the same time it is more clear to indicate which effects were statistically confirmed and which were not.
Response 10: According to the reviewer 's opinion, the explanation part of each paragraph of the result analysis part has been deleted, Lines 196-199, 203-202, 292-296, 300-303 305-307, 311-312,323-325, 327-329, 459-461.
Point 11:L373: The Discussion is too short. Please discuss in more detail the possible mechanisms of influence of phosphorus availability on the studied indicators. It is also worth not only pointing out that your results contradict some previously published data, but also trying to explain why this happened.
Response 11: Thanks to the reviewer's opinion, we have revised the discussion as required, lines 495-504.
Point 12:Please carefully check the text of the manuscript (many typos). If possible, I would advise showing the manuscript to an English-speaking person before resubmitting.
Response 12: Thanks to the opinion of the reviewer, we have carefully checked the manuscript.
Round 2
Reviewer 2 Report
The majority of my comments was taken into account during revision. I tend to think that the manuscript may be published in present form.